# Identification of a conserved virion-stabilizing network inside the interprotomer pocket of enteroviruses

Justin W. Flatt [1,2], Aušra Domanska [1,2], Alma L. Seppälä [1,2] & Sarah J. Butcher [1,2✉]

Enteroviruses pose a persistent and widespread threat to human physical health, with no specific treatments available. Small molecule capsid binders have the potential to be developed as antivirals that prevent virus attachment and entry into host cells. To aid with broad-range drug development, we report here structures of coxsackieviruses B3 and B4 bound to different interprotomer-targeting capsid binders using single-particle cryo-EM. The EM density maps are beyond 3 Å resolution, providing detailed information about interactions in the ligand-binding pocket. Comparative analysis revealed the residues that form a conserved virion-stabilizing network at the interprotomer site, and showed the small molecule properties that allow anchoring in the pocket to inhibit virus disassembly.

[1] Faculty of Biological and Environmental Sciences, Molecular and Integrative Bioscience Research Programme, University of Helsinki, Helsinki, Finland.
[2] Helsinki Institute of Life Sciences, Institute of Biotechnology, University of Helsinki, Helsinki, Finland. ✉email: sarah.butcher@helsinki.fi

The group B coxsackieviruses (CVBs) are a major source of both acute and chronic diseases in humans. Age and immune status are thought to be the main determinants of morbidity and mortality, with infants, young children, and immunocompromised individuals being particularly susceptible to serious and sometimes life-threatening infections. Coxsackievirus B3 (CVB3) can cause cardiac arrhythmias and acute heart failure[1,2]. Additionally, CVB3 infections during pregnancy have been linked to an increase in neurodevelopmental delays, fetal myocarditis, and spontaneous abortions[3,4]. Coxsackievirus B4 (CVB4) appears to elicit or enhance certain autoimmune disorders such as type 1 diabetes as the virus has been isolated from individuals diagnosed with rapid onset type 1 diabetes, and these isolates were then shown to cause diabetes in mice models[5,6]. Dotta et al.[7] have provided arguably the most direct support for CVB4 as a viral trigger of diabetes via immunohistochemical detection and sequencing of virus from the pancreatic tissue of diabetic patients. Thus, it is of great importance to develop antiviral drugs and vaccines to combat CVBs, as well as other enteroviruses, given that cases and outbreaks can result in substantial hospitalization and burden of healthcare services.

CVB capsids share a common enteroviral architecture constructed from 60 repeating asymmetric units termed protomers, each consisting of the four structural proteins VP1, VP2, VP3, and VP4 (ref. [8]). The protomers assemble to form the ~30 nm wide icosahedral shell with a pseudo $T = 3$ arrangement that encapsidates the linear single-stranded RNA genome. The arrangement occurs because of the similar structures of VP1, VP2, and VP3, which all adopt an eight-stranded, antiparallel β-barrel fold despite having low sequence homology. The four strands of the β-sheets are connected by hypervariable loops that are responsible for the high antigenic diversity of enteroviruses. The organization of the 180 β-barrels is much the same as observed in $T = 3$ lattices formed by 180 identical copies of a capsid protein, with VP1 localized to fivefolds, while VP2 and VP3 alternate around the two- and threefold axes. VP4 is located on the inside of the capsid and is myristoylated. Many picornaviruses utilize a canyon-like feature on their surface to bind cellular receptors belonging to the immunoglobulin superfamily[9]. Binding into the canyon destabilizes virions and initiates the uncoating process by triggering release of the lipid moiety "pocket factor" from the small hydrophobic pocket in VP1 (ref. [10]). Notable exceptions include rhinovirus C and parechoviruses, which do not accommodate a fatty-acid pocket factor[11–13].

Small molecules that bind tightly and specifically to conserved capsid features to interfere with virus entry or uncoating are among the most promising strategies for blocking enterovirus infections[14]. These molecules, the WIN antiviral compounds, target the VP1 hydrophobic pocket, which has an entrance located at the base of the canyon-like depression surrounding each capsid fivefold axis[15]. The site is normally occupied by the pocket factor; however, binding of chemically optimized compounds dislodges the lipid due to the drugs having a much higher binding affinity[16]. Replacement of the pocket factor with capsid binders provides entropic stabilization by raising the uncoating free energy barrier against thermal or receptor-induced conformational changes[17,18]. In this way, the compounds are able to prevent formation of expanded 135S intermediates or A-particles, which is a required step for genome release. In vitro testing has shown this to be the case for several VP1 pocket binders; they possess high potency and broad-spectrum activity against enteroviruses. However, clinical development has been thwarted because of issues related to efficacy and toxicity, as well as emergence of drug-resistant viruses[19,20]. Recently, we discovered a second druggable pocket at a conserved VP1–VP3 interprotomer interface in the viral capsid[21]. This interface is in a region of the capsid that undergoes quaternary conformational changes to promote disassembly and release of the virion's genome into the host cell. Synthetic compounds that occupy the interprotomer pocket are inhibitors of a large number of enteroviruses, and act synergistically with inhibitors that target the VP1 pocket.

Here, in an effort to better understand the druggable features of the interprotomer pocket, we have analyzed high-resolution structures of two medically important enteroviruses, coxsackieviruses B3 and B4, complexed with interprotomer-targeting compounds CP17 and CP48, respectively. The structures were determined by cryo-electron microscopy (cryo-EM) to beyond 3 Å resolution, which allowed us to identify the detailed interactions that facilitate drug binding at the VP1–VP3 interface. In addition to modeling the key residues, we also calculated interaction energies for both compounds using in silico methods. We found that both compounds target the same interprotomer side chains, and the energy of interaction is comparable to what has been observed for robust, high-affinity binders of the VP1 hydrophobic pocket. These results taken together help to explain how this new class of drugs interferes with virus uncoating, and indicate that it is worthwhile to focus on developing therapies that include a synergistic combination of binders to potentially improve efficacy, alleviate side effects, and shorten treatment of enteroviral infections.

## Results and discussion

**CP17 bound to the interprotomer pocket of CVB3**. CP17 is a benzenesulfanomide derivative that potently inhibits the CVB3 Nancy strain in cells (EC$_{50}$ $0.7 \pm 0.1$ μM) via a direct interaction with the capsid that increases virion thermostability by 1.5 and 2.1 log$_{10}$ TCID$_{50}$/mL at 46 and 49 °C, respectively[21]. A 4.0 Å cryo-EM structure of CP17 in complex with CVB3 Nancy (EMD-0103) revealed that the site of binding is located at a conserved VP1–VP3 interprotomer interface, but the low resolution of the map prevented identification of the detailed interactions within the pocket. We reprocessed the raw data (EMPIAR-10199) using RELION 3.0 and the resolution improved to 2.8 Å (Fig. 1a)[22]. The cryo-EM map shows pronounced backbone features for the four structural proteins of CVB3, and well-defined density for CP17 on the surface of the capsid (Fig. 1b–d). Importantly, the resolution is now sufficient for describing specific ligand–protein interactions (Fig. 1c). The interprotomer site is located between adjacent asymmetric units "protomers" in a narrow opening formed at the intersection of neighboring VP1 β-barrels and the C terminus of a proximally situated VP3 molecule. Three residues that play a key role in binding CP17 are conserved across enteroviruses. In CVB3 Nancy they are Arg219 (VP1), Arg234 (VP1), and Gln233 (VP3) (Fig. 1c). The Arg residues, which come from neighboring VP1 polypeptide chains, are situated in the deepest part of the pocket, and in the high-resolution structure, we observed that their guanidinium groups form salt bridges with the carboxylic end of CP17 (Fig. 1e). The Gln residue from the C terminus of VP3 is positioned at the entrance of the drug site, where the oxygen in the side chain engages in a hydrogen bond interaction with the NH located in the elbow region of the inhibitor. In addition, there are other contributions inside the pocket such as a cysteine residue in VP1 (C73) and hydrophobic residues (VP1 F76 and VP3 F236) that interact with the benzene rings of the compound, contributing to the overall binding energy and specificity of CP17 (Fig. 1c, e). The binding energy (sum of van der Waals and electrostatics) for CP17 is −74 kcal/mol based on the NAMD energy plugin in VMD. The predicted value is comparable to potent inhibitors of the VP1 hydrophobic pocket, namely the capsid binders GPP3 (−66 kcal/mol) and NLD (either −69 or −64 kcal/mol depending on protonation state)[23].

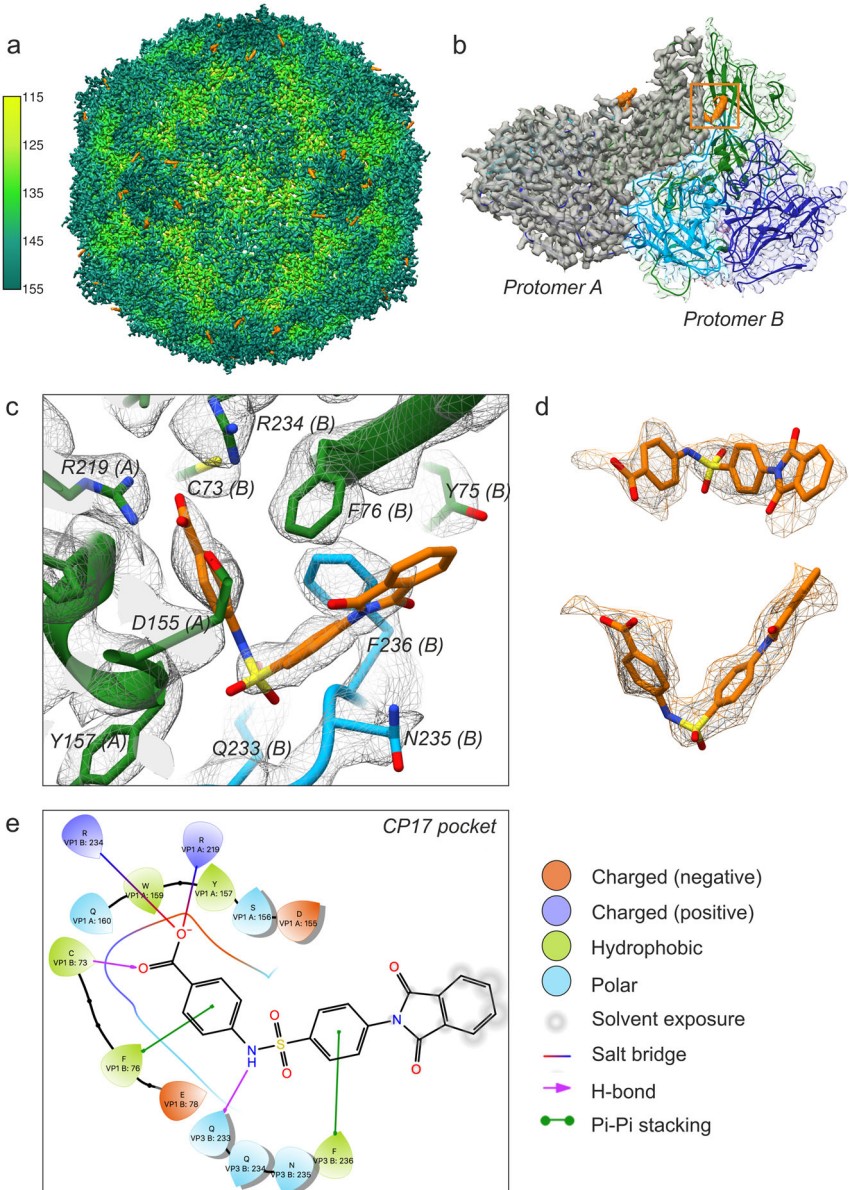

**Fig. 1 Cryo-EM structure of CP17-bound CVB3. a** Three-dimensional reconstruction of CVB3 after incubation with a saturating amount of capsid binder. The virion is viewed along the icosahedral twofold axis and colored according to radial distance in Å from the particle center. Density for CP17 is shown in orange. The map was resolved to 2.8 Å upon reprocessing raw data from a previous publication (see ref. [19]; EMPIAR-10199). **b** CP17 binds in a pocket between neighboring protomers. VP1, green; VP2, dark blue; VP3, light blue. **c** Inhibitor and pocket residues at a display contour level of 2.5σ (σ is the standard deviation of the density map). **d** CP17 shown in density contoured to 1.6σ. **e** Ligand interactions diagram for CP17 generated by Schrödinger Maestro v12.02.

**CP48 within the CVB4 virion**. To further investigate and confirm the activity and structural basis for how inhibitors bind at the interprotomer pocket, we determined a structure of CVB4 in the presence of a commercially available analog, which we refer to as CP48. This particular inhibitor was found to be active against all six serotypes of CVBs, and completely inhibited poliovirus type 1 replication at a concentration of 144 μM[21]. We chose to work with CVB4 because no structure exists for this important human pathogen. First, we confirmed that addition of CP48 increases CVB4 thermal stability (Fig. 2 and Supplementary Data). Then, purified virus was incubated with a saturating concentration of CP48 (virus:drug molar ratio of 1:2500), applied to grids, and flash-frozen for cryo-EM. After image processing, a subset of 18,626 particles yielded a 2.7 Å reconstruction using the FSC 0.143 threshold criterion[24]. The outer surface of the virus

particle is similar to that of other enteroviral capsids, with major features including the fivefold star-shaped mesas, threefold propeller-like protrusions, and twofold depressions (Fig. 3a). In addition, there is stable and well-defined density for the inhibitor at the interprotomer site (Fig. 3a, c, d). The control structure of CVB4 incubated without compound revealed no additional density inside the pocket (Fig. 3b, e). We did not detect a conformational change induced by the presence of CP48 (RMSD for native versus CP48-bound virus: 0.45 Å). Modeling confirmed the critical role of the conserved pocket side chains: two Arg residues on the inner surface and a Gln residue at the entrance (Fig. 3d). Similar to how CP17 is anchored to CVB3, CP48 is stabilized by stacking interactions with two hydrophobic residues, Y67 in VP1 and F236 in VP3 (Fig. 3d, f). The C64 residue in VP1, though not involved in the direct interaction with CP48, is in close vicinity to

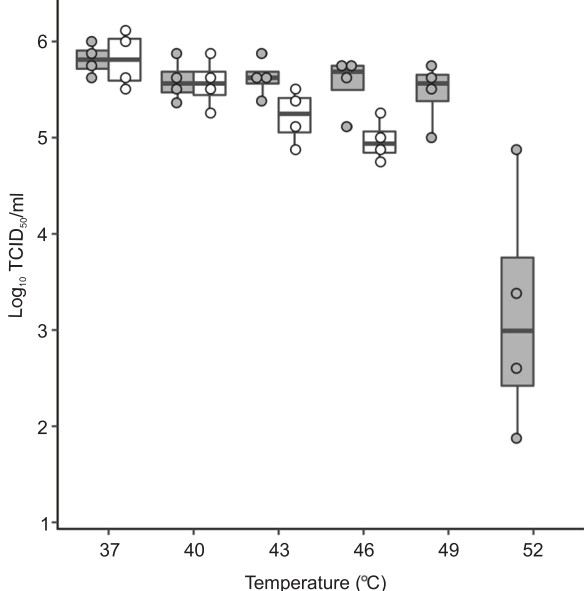

**Fig. 2 A box and whisker plot showing the effect of CP48 on CVB4 thermal stability.** Thermostability assay in the presence (gray) or absence (white) of interpromoter-targeting CP48. No virus was detected in control samples at 49 and 52 °C at the $TCID_{50}$/mL detection limit of 2 $log_{10}$. $N = 4$ independent experiments, the middle bar is the median, the boxes represent quartile data distribution, and individual data points are shown as circles. Figure generated with the command ggplot in R.

the compound (Fig. 3f). The energy for the bound inhibitor is approximately −45 versus −74 kcal/mol for CP17–CVB3, while the $EC_{50}$ for CP48–CVB4 is only 8.6 ± 0.8 μM compared to an $EC_{50}$ of 0.7 ± 0.1 μM for CP17–CVB3 (ref. [21]).

There were other notable differences in the CVB4 structure. First, in CP48–CVB4, density corresponding to the native pocket factor within the β-barrel of VP1 appears to be altered by the presence of drug (Supplementary Fig. 1a, b); however, there were no major conformational changes in the four capsid proteins based on comparison to the control structure of CVB4 alone (Supplementary Fig. 1c). The alteration may result from a change in the structural dynamics of the pocket factor, if this region is an additional weak binding site for CP48, or it may reflect a substoichiometric occupancy issue where signal is averaged away by applying icosahedral symmetry during image processing. Worth noting, the CVB3 Nancy strain lacks density for the lipid factor (Supplementary Fig. 1b). The Nancy strain has a substitution of Leu for Ile at position 92 in the hydrophobic pocket of VP1, which correlates with resistance to pleconaril-like compounds. Experiments involving a pleconaril-sensitive CVB3 Nancy variant (VP1 L92I) showed that when pleconaril and CP17 are combined, they have a synergistic effect, which suggests that the druggable site in VP1 does not contribute to the antiviral mechanism of interpromoter-targeting compounds[21]. Another difference between CVB4 and CVB3 occurs in the exposed loop linking β-strands B and C of VP1, which for CVB4 differs in length and conformation (Supplementary Fig. 1d). This loop is known to be a serotype-specific-neutralizing antigenic site[25].

**The virion-stabilizing network inside the pocket.** Precise mechanistic descriptions of how capsid binders target enteroviruses has many practical applications, in terms of both understanding the cell biology of virus entry and design of new therapeutic agents. However, the large, flat, and relatively featureless surface of enterovirus capsids poses many challenges for

drug design. Here, we used high-resolution cryo-EM to reveal how a new class of capsid binders makes stabilizing contacts inside the interprotomer pocket.

The network comprises about 15 residues with 3 highly conserved amino acids forming the core of the binding site. These three residues, each from a different polypeptide chain, dictate the size and shape of interprotomer-targeting compounds, as well as the mechanism of action (Fig. 4a). Other interactions that define the structure–activity relationship within the pocket include a pair of hydrophobic residues that stably position the benzene scaffolds of the inhibitors, and a cysteine residue that can hydrogen bond with the carboxylic end (Fig. 4b, c). When an inhibitor is stably anchored to the network (60 sites per a single virion), it interferes with motion transmission such that the virus particle cannot undergo expansive conformational changes in the interprotomer region, and hence is unable to uncoat the genome at precisely the right time in infection.

**Conclusion**

Despite decades of research on WIN antiviral compounds with, e.g., $EC_{50}$ value for pleconaril against human rhinovirus B1 reported as 0.2 μM, no drugs have been approved for use against enteroviruses[26]. Recently, a new class of broad-spectrum capsid binders was described, which inhibit a variety of enteroviruses by occupying a positively charged surface depression in the interprotomer zone with, e.g., $EC_{50}$ value for CP17 against CVB3 as 0.7 μM. Structure-guided in vitro assays involving CVB3 and CP17 indicated that this class of capsid binders increases particle stability, which we have observed to be the case here for CVB4 and CP48. Virus variants with reduced susceptibilities to compounds targeting either pocket can be selected under pressure, with concomitant reduced viability[21,27]. Reverse engineering mutation experiments revealed that four interprotomer mutants in CVB3 Nancy were not viable: VP1 Q160G, VP1 R234G, VP3 F236G, and VP3 Q233G (VP1 R219 was not tested)[21]. We were unable to perform similar experiments with CVB4 here because there is no infectious clone available. Nevertheless, structural alignments and experimental data suggest a conserved virion-stabilizing network within the interprotomer pocket that is less tolerant to mutations, a promising result for efforts to develop antivirals. Interestingly, Duyvesteyn et al.[28] recently published a 1.8 Å resolution X-ray structure of bovine enterovirus F3 (EV-F3) with glutathione (GSH) positioned in a similar way within the pocket. The antioxidant engages the same virion-stabilizing network as CP17 and CP48, which is not surprising given that these molecules have strikingly similar geometrical and chemical features (Supplementary Fig. 2). Specifically, GSH adopts a hook-shaped structure with a carboxylic end and a sulfur-containing elbow region. The overall size approximates that of inhibitors that occupy the interprotomer site. The carboxylic group interacts with the two Arg residues from neighboring VP1 polypeptides inside the pocket, while the sulfur atom interacts with the oxygen in the Gln side chain of VP3. It is believed that for CVB3 and CVB4, GSH makes strong interactions with adjacent protomers to facilitate intracellular assembly of progeny virions[29–31]. Further work is necessary to understand how these molecules, which share similar shape and chemistry, modulate stability at the pocket to either prevent uncoating or facilitate assembly. Outside of structural efforts, it will be important to assess the influence of cellular cues and factors on their different modes of action. Accordingly, we foresee the development of improved binders, as well as an enhanced understanding of the biological significance of the interprotomer site, by building on the information provided by these new structures which show molecules inside the pocket.

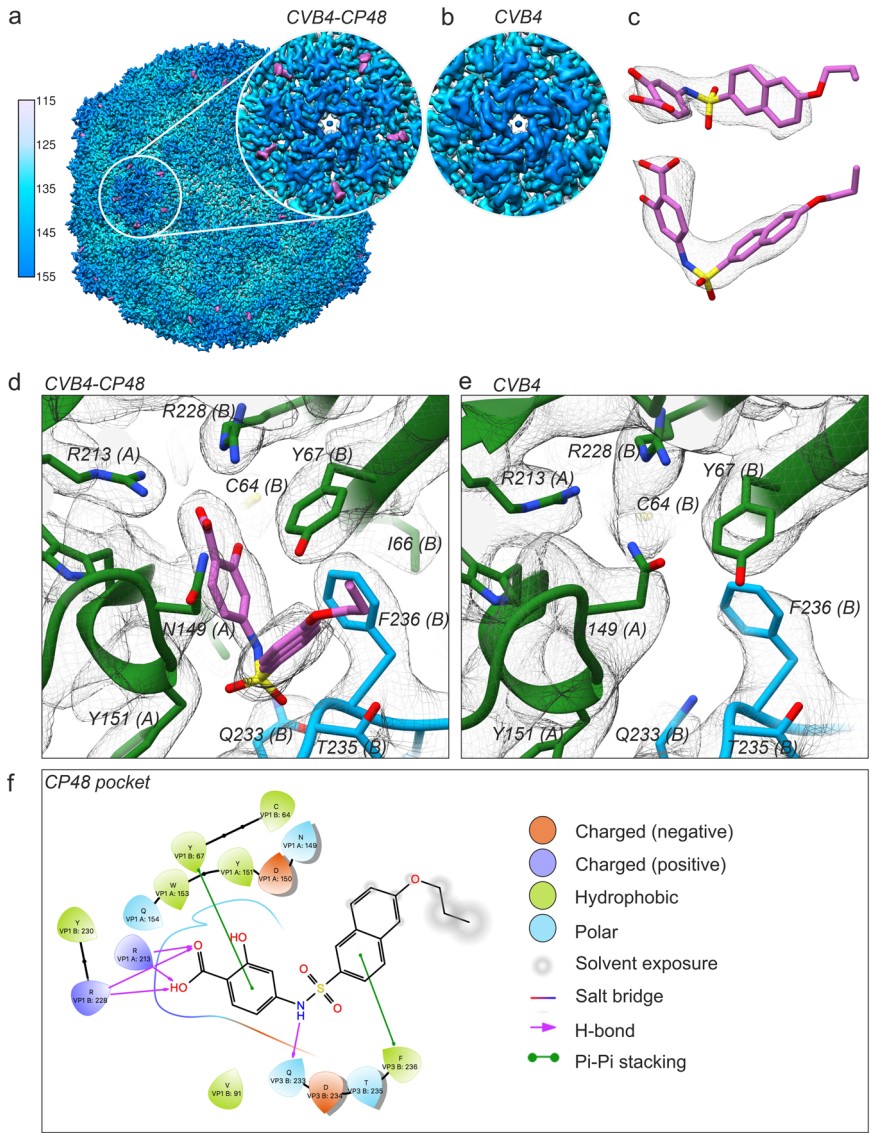

**Fig. 3 Cryo-EM of the CP48–CVB4 complex in comparison to CVB4 alone. a** Visualization of CVB4 in the presence of CP48. The view is along the twofold axis with radial coloring, and the inset shows clear inhibitor density at the interprotomer site near the fivefold axis of symmetry. Density for the 385.44 Da CP48 is displayed in magenta. **b** The corresponding region of the cryo-EM density of the CVB4 control. **c** CP48 fits well into the additional density detected in the cryo-EM map of the CP48–CVB4 complex. **d** A close-up view of the capsid binder (magenta) within the interprotomer pocket. **e** A close-up view of the interprotomer pocket in the cryo-EM map of CVB4 alone shows no density for the compound. **d**, **e** Color coding for the viral proteins VP1, VP2, and VP3 is the same as in Fig. 1. **f** Interaction diagram of CP48 with CVB4 viral proteins generated in Schrödinger Maestro software v12.02.

## Methods

**Virus culture and purification**. BGM cells were a kind gift from the Rega Institute for Medical Research in Leuven. No authentication of the cell line was done. Cell supernatant was routinely tested as mycoplasma negative using the Eurofins Genomics Company mycoplasma testing service. Cells were cultured in Eagle's minimum essential medium (MEM) supplemented with 10% fetal bovine serum (FBS), 1× nonessential amino acids, 1% GlutaMAX, and 1% antibiotic–antimycotic solution in a chamber environment adjusted to 37 °C and 5% $CO_2$. To produce virus particles for the study, 30 confluent T175 flask were inoculated with CVB4 (GenBank: AF311939.1) at a multiplicity of infection of ~0.5 in serum-free medium. Additionally, the infection medium contained 20 mM HEPES (pH 7.0). At 3 days post-infection, widespread viral cytopathic effect was evident, and the contents of each flask were collected, freeze–thawed three times, and centrifuged at 4000 r.p.m. and 4 °C for 10 min to remove cellular debris. The supernatant was then carefully removed and concentrated using a Centricon centrifugal filter device (100 kDa cut-off). Virus particles were purified by centrifuging through a CsCl gradient (top density 1.25 g/cm³ and bottom density 1.48 g/cm³) at 30,000 r.p.m. and 4 °C for 19 h. The gradient/exchange buffer consisted of 10 mM HEPES (pH 7.0), 150 mM NaCl, 2 mM $MgCl_2$, and 2 mM $CaCl_2$. Bands containing intact virions were collected and the CsCl was removed by buffer exchange.

**Thermostability assay**. Approximately $5 \times 10^4$ TCID$_{50}$ units of CVB4 strain E2 was mixed with 20 μM concentration of CP48 in six tubes (reaction volume 52 μL) and incubated over a range from 37 to 52 °C for 2 min, followed by rapid cooling on ice. The infectious virus load in the samples was estimated by an end-point titration assay using BGM cells. Specifically, serial 10 log dilutions were prepared in infection medium and applied to BGM cell monolayers on 96-well plates arranged one day before use by seeding $2 \times 10^4$ cells per well. Two days after infection, the BGM cell monolayers were examined for cytopathic effects and TCID$_{50}$/mL was estimated using Kärber–Spearman formula[32]. The experiment was repeated with four independent biological replicates for each measurement. The detection limit is $10^2$ TCID$_{50}$/mL.

**Cryo-EM sample preparation and data collection**. The compound 4-{[(6-propoxy-2-naphthyl)sulfonyl]amino}benzoic acid (CP48) was ordered from a commercial supplier (www.specs.net) and dissolved in DMSO at a concentration of 10 mg/mL. We further diluted (10× dilution) the compound in a gradient/exchange buffer. Purified CVB4 and CP48 were then mixed at a molar ratio of 1:2500, which yielded a final capsid binder concentration of 0.17 mg/mL. The mixture was incubated at 37 °C for 1 h. For cryo-EM sample preparation, 3.0 μL samples of CVB4–CP48 were applied to glow-discharged grids (Ted Pella product No. 01824).

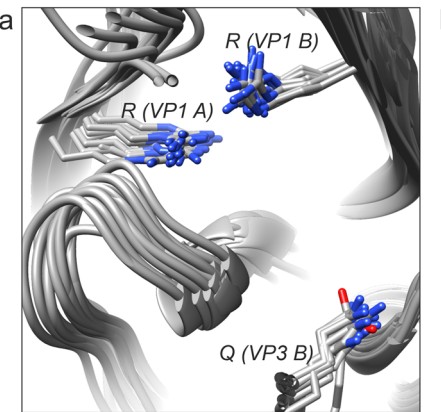
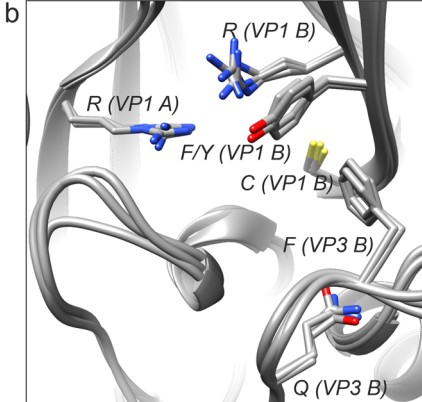

c

|  | | Protomer A | Protomer B | | | | |
|---|---|---|---|---|---|---|---|
| *Enterovirus* | | VP1 | VP1 | VP3 | VP1 | VP1 | VP3 |
| A | CVA16 (PDB 5C4W) | **R236** | **R250** | **Q237** | L89 | I92 | N240 |
|  | EVA71 (PDB 3VBS) | **R236** | **R250** | **Q237** | L89 | E92 | T240 |
| B | CVB3-CP17 (PDB 6ZCL) | **R219** | **R234** | **Q233** | C73 | F76 | F236 |
|  | CVB4-CP48 (PDB 6ZCK) | **R213** | **R228** | **Q233** | C64 | Y67 | F236 |
|  | CVA9 (PDB 1D4M) | **R222** | **R237** | **Q233** | C73 | M76 | K236 |
|  | E1 (PDB 1EV1) | **R222** | **R237** | **Q234** | C73 | Y76 | F237 |
|  | E7 (PDB 2X5I) | **R222** | **R237** | **Q233** | C73 | I76 | L236 |
|  | E11 (PDB 1H8T) | **R222** | **R237** | **Q233** | C73 | M76 | L236 |
| C | PV1 (PDB 1HXS) | **R243** | **R258** | **Q233** | C86 | I89 | * |
| D | EVD68 (PDB 4WM8) | **R223** | **R238** | **Q235** | L68 | L72 | H238 |
| F | EVF3-GSH (PDB 6T4C) | **R216** | **R229** | **Q238** | L74 | M78 | A241 |
| *Rhinovirus* | | | | | | | |
| A | RVA2 (PDB 1FPN) | **R219** | **R234** | **Q231** | C76 | E79 | A243 |
| B | RVB14 (PDB 4RHV) | **R227** | **R242** | **Q230** | C76 | V79 | A233 |
| C | RVC15 (PDB 6JZG) | **R210** | K221 | **Q230** | L79 | K82 | E233 |

**Fig. 4 Stabilization inside the interprotomer pocket of enteroviruses. a** Structural alignment of the three key residues that form the core of the virion-stabilizing network inside the interprotomer pocket. Atomic models used for the alignment are listed in C. **b** Alignment of PDB IDs 6ZCK, 6ZCL, and 1EV1 with the same view as **a** but with CP17/48-stabilizing hydrophobic residues of the interprotomer pocket added to the visual. **c** Conservation of anchor residues in the pocket based on the structural data presented in this study. The three residues that are highly conserved are in bold whereas the other major elements (hydrophobic and cysteine) vary. *Enterovirus A, B, C, D,* and *F* and *Rhinovirus A, B,* and *C* species are indicated in column 1 along with the wwPDB IDs. The numbering of the residues in columns 2–6 were taken from the wwPDB files listed in column 1. (*) not modeled in the coordinates for 1HXS.

Grids were manually blotted with filter paper to remove excess sample and flash-frozen in liquid ethane with a homemade plunger. In all, 300 kV data acquisition was carried out at the Science for Life Laboratory, Stockholm, Sweden (Table 1). The frozen-hydrated grids were loaded into a FEI Titan Krios electron microscope operated at 300 kV. A total of 4379 movies were acquired with a Gatan K2 Summit direct electron detection camera at a nominal magnification of ×130,000, giving a pixel size of 1.06 Å per pixel. The total electron dose was approximately 46 electrons per $Å^2$ fractionated into 30 frames. Frame images in each movie were aligned and averaged to correct for beam-induced motion using MotionCor2 (ref. [33]). A control dataset of CVB4 without CP48 was collected within the Instruct-ERIC Center Finland at the University of Helsinki using a Talos Arctica equipped with a Falcon III direct electron detection camera (Table 1). A total of 8860 movies were acquired at a nominal magnification of ×120,000, giving a pixel size of 1.24 Å per pixel. The accumulated electron dose was approximately 30 electrons per $Å^2$ fractionated into 30 frames. MotionCor2 was used to produce a single micrograph from aligned and averaged movie frames (Supplementary Fig. 3).

**Image processing.** Defocus values of CVB4–CP48 micrographs were determined by Gctf[34]. A total of 31,436 particles were picked from 4377 micrographs using ETHAN[35]. Orientation and center parameters were determined and refined using RELION-3 within the Scipion image processing framework[22,36]. Reference-free two-dimensional classification was used to discard 12,310 particles in poorly defined classes or false positives. An ab initio model generated with the RELION 3D Initial Model protocol was used as an initial reference model for maximum-likelihood three-dimensional classification. One class containing 18,626 high-quality particles was selected and divided into random halves for further refinement. The initial round of refinement was followed by subsequent rounds that included iterative per-particle CTF-refinement and Bayesian polishing. After convergence in the final refinement step, and FSC curve was calculated and the resolution was determined to be 2.7 Å according to the gold-standard FSC = 0.143 threshold criterion. A *B*-factor of −70 $Å^2$ was applied to sharpen the density map for modeling and analysis. The same image processing approach was applied to CVB4 without drug, as well as the raw data for CVB3–CP17 (EMPIAR-10199), which resulted in 3.4 and 2.8 Å maps, respectively.

ResMap images, map cross-sections, and FSC curves for the three structures are included in Supplementary Fig. 4 (ref. [37]).

**Modeling.** An initial template for CVB4 capsid proteins VP1-VP4 was derived from a homology-based model calculated by I-TASSER[38]. The UCSF Chimera Build Structure tool was used to translate the Simplified Molecular Input Line Entry Specification (SMILES) string for CP48 into a three-dimensional structure and parameterization was completed using SwissParam[39,40]. Structures for viral proteins and drug were docked into the EM density using UCSF Chimera, followed by iterative manual adjustment and real-space refinement using COOT[41]. Sequence assignment was guided by bulky amino acid residues such as Phe, Tyr, Trp, and Arg, and featureful density allowed placement of the ligand. The optimized model for CVB4–CP48 was then subjected to end-stage refinement using the molecular dynamics flexible fitting program originally developed by Klaus Schulten and co-workers[42]. Harmonic restraints were applied to prevent overfitting during simulations. Capsid proteins for the CVB4 virion without drug were modeled using a similar protocol and comparison to the CP48–CVB4 structure confirmed that drug binding does not induce conformational changes in the virion. The RMSD between CP48–CVB4 and CVB4 alone was 0.45. The binding energy for CP48 inside the interprotomer pocket was obtained using the NAMD energy plugin in VMD[43,44]. We used the same procedure to refine atomic coordinates for CVB3–CP17 (PDB ID code 6GZV) into the newly determined CVB3–CP17 2.8 Å map. The structural alignment of PDB files was done using the MatchMaker feature of UCSF Chimera. Ligand Interactions diagrams for the compounds in the interprotomer pockets were generated by Schrödinger Maestro v12.02 (Shrödinger Release 2019-4: Maestro v12.2, Schrödinger, LLC, New York, NY, 2020).

**Statistics and reproducibility.** The thermostability assay was analyzed as a box and whisker plot from *n* = 4 independent experiments. The summary of the cryo-EM data collection, refinement, and validation statistics are shown in Table 1 (ref. [45]).

**Table 1 Cryo-EM data collection, refinement, and validation statistics.**

| | CVB4–CP48 (EMD-11165) (PDB 6ZCK) | CVB3–CP17 (EMD-11166) (PDB 6ZCL) (EMPIAR-10199 (ref. [21])) | CVB4 (EMD-11300) (PDB 6ZMS) |
|---|---|---|---|
| *Data collection and processing* | | | |
| Magnification | 130,000 | 130,000 | 120,000 |
| Voltage (kV) | 300 | 300 | 200 |
| Electron exposure ($e^-$/Å$^2$) | 47 | 47 | 30 |
| Defocus range settings (µm) | −0.6 to −3.0 | −0.6 to −3.0 | −0.1 to −2.0 |
| Pixel size (Å) | 1.06 | 1.06 | 1.24 |
| Symmetry imposed | I2 | I2 | I2 |
| Initial particle images (no.) | 31,436 | 17,300 | 96,985 |
| Final particle images (no.) | 18,626 | 13,252 | 40,627 |
| Map resolution (Å) | 2.7 | 2.8 | 3.4 |
| FSC threshold | 0.143 | 0.143 | 0.143 |
| Map resolution range (Å) | 999–2.12 | 999–2.12 | 999–2.48 |
| *Refinement* | | | |
| Map sharpening B factor (Å$^2$) | −70 | −77 | −90 |
| Model composition | | | |
| Non-hydrogen atoms | 6443 | 6370 | 6397 |
| Protein residues | 800 | 798 | 800 |
| Ligands | 2 | 2 | 0 |
| R.m.s. deviations | | | |
| Bond lengths (Å) | 0.91 | 0.86 | 0.90 |
| Bond angles (°) | 1.13 | 1.01 | 1.01 |
| Validation | | | |
| MolProbity score | 1.4 | 1.16 | 1.52 |
| Clashscore | 0 | 0 | 0 |
| Poor rotamers (%) | 3.9 | 2.4 | 1.7 |
| Ramachandran plot | | | |
| Favored (%) | 93 | 94 | 91 |
| Allowed (%) | 5 | 5 | 6 |
| Disallowed (%) | 2 | 1 | 3 |

**Reporting summary**. Further information on research design is available in the Nature Research Reporting Summary linked to this article.

## Data availability

The datasets generated during and/or analyzed in the current study are available in the wwPDB repositories with the persistent web links: https://doi.org/10.2210/pdb6ZCL/pdb; https://doi.org/10.2210/pdb6ZMS/pdb; and https://doi.org/10.2210/pdb6ZCK/pdb and the Electron Microscopy Pilot Image Archive with the persistent web link https://doi.org/10.6019/EMPIAR-10199. The accession numbers are: CVB3–CP17: PDB 6ZCL, EMPIAR-10199 (ref. [21]) and EMD-11166, CVB4: PDB 6ZMS and EMD-11300 and CVB4–CP48: PDB 6ZCK and EMD-11165. The source data underlying Fig. 2 are provided in Supplementary Data.

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

## Acknowledgements

We thank Benita Löflund and Pasi Laurinmäki of Instruct-ERIC Centre Finland and the Biocenter Finland National Cryo-Electron Microscopy Unit, Helsinki University; Marta Carroni at the CryoEM Swedish National Facility funded by the Knut and Alice Wallenberg, Family Erling Persson and Kempe Foundations, SciLifeLab, Stockholm University and Umeå University, and the CSC-IT Center for Science Ltd for providing technical assistance and facilities to carry out the work. Additionally, we thank Johan Neyts and the Rega Institute for Medical Research in Leuven for kindly providing the BGM cell line and the CVB4 strain E2 used in this study. This project was supported by the Academy of Finland (315950), the Sigrid Juselius Foundation, the People Program (Marie Curie Actions) of the European Union's Seventh Framework Program FP7/2007-2013/ under REA grant agreement number 612308, and the European Union's Horizon 2020 Research and Innovation Program under the grant agreement number 857203.

## Author contributions

J.W.F., A.D., and S.J.B. conceived the idea and designed the experiments. J.W.F., A.D., and A.L.S. carried out the experiments. J.W.F., A.D., A.L.S., and S.J.B. contributed to interpretation of results and writing of the manuscript.

## Competing interests

The authors declare no competing interests.
