## [Peer Review File · Communications Biology]

Reviewers' comments:

Reviewer #1 (Remarks to the Author):

Review for COMMSBIO-20-2076-T

This manuscript by Flatt et al, "Identification of a conserved virion-stabilizing network inside the interprotomer pocket of enteroviruses" reports on two cryo-EM derived structures of icosahedral viruses bound to drug compounds. Enteroviruses are important human pathogens and an atomic level understanding of how drug compounds interact with viral proteins is an important area of research.

In this work, the authors reprocess old data (formerly a 4 Å map) to generate a map closer to 2.8 Å, allowing them to fully model the proteins and the bound ligand. In addition, they solved a novel structure to 2.7 Å. The work is largely structural, although they also do thermal titrations to show that the virions are stabilized against heat when bound to the compounds. I have very few issues with the quality of the data or the authors' main conclusions. The cryo-EM data collection and interpretation section is high quality, and is thorough. However, there are several points missing throughout the manuscript, and I have a number of suggestions regarding the presentation that would make this manuscript easier to digest.

The Introduction is generally well written and comprehensive. The term "protomer" could be confusing to non-experts and I suggest providing a better description of this term.

Suggestions on presentation:

The figures are hard to interpret. For example:

Figure 1: An enlarged view of the binding pocket with residue labels would help the reader immensely. Perhaps a "cage view" of the density would also help.

Figure 2: I assume the control was done at 49 and 52 and no active virus was detected? Can this be shown more clearly with an asterisk or a "ND" label for "not detected"?

Figure 3C: superimposing the two structures with different colors would help interpret the similarities and differences better. And the pink density blob is not particularly effective in showing the point. Lastly, a view showing residue labels would help quite a bit (same as for figure 1).

Figure 4A: label the residues and/or add arrows. It is really hard to see what the authors are trying to highlight.

Table of cryo-EM data: Was a defocus value of -5 μm really used? Seems rather unconventional. Also, please show a raw micrograph of the new data (supplementary is fine).

General comments on drug effectiveness:

Interpretation on page 9: "Also, the control CVB4 structure has no density filling the interprotomer pocket. The alteration may result from a change in the structural dynamics of the pocket factor, or because this region is an additional weak binding site for CP48".....These structures were solved using icosahedral symmetry, so there could also be issues with stoichiometry and occupancy. If all 60 sites are not occupied, the signal would simply be averaged out.

Related to the point above, on page 11, the authors state "When an inhibitor is stably anchored to the network (60 sites per a single virion), it interferes with motion transmission such that the virus particle cannot undergo expansive conformational changes in the interprotomer region, and hence is unable to uncoat the genome at precisely the right time in infection.".... Is it known if full occupancy is required for the drug to function? An experiment doing a titration of the compound

along with virus infectivity assays. If this is known, please cite the previous work.

In the materials and methods the drug compounds were incubated with virus at 10mg/mL. This seems a very high concentration and I wonder how tight the binding really is. Is this dose toxic to humans? See above comment on titration... if this study really is evaluating the binding potential of these viruses, the nature of the binding really should be explored further. As one example, comprehensive site directed mutagenesis would be a logical follow up to determine which of the residues are critical for drug binding.

Reviewer #2 (Remarks to the Author):

This is a follow-up report of a recent article on interprotomer-targeting compounds to enterovirus by the authors (Abdelnabi et al., 2019, Plos Biol). Novel information provided in current study is 1) a high resolution structure of a CP17-CVB3 complex, and 2) a novel structure of a CP47-CVB4 complex. These structures revealed conserved three amino acid residues of enterovirus capsid proteins interacting with interprotomer-targeting compounds.

The direct importance suggested by this manuscript might be as follows:

1. Mode of interaction between interprotomer-targeting compounds and critical amino acid residues (two Arg residues and one Gln residue), which is similar to a glutathione-EVF3 complex (Duyvesteyn et al., 2020, Commun Biol).
2. Atomic structure of CVB4.

A number of structures of enterovirus have been solved by atomic level, including the structures of interprotomer pocket. With identified critical amino acid residues, structure-guided drug design might provide more potent interprotomer-targeting compounds.

Atomic structure of CVB4 could provide a structural basis for understanding of CVB4 infection and for development of potent inhibitors.

Some parts of this manuscript might be clarified:

1. Apparently, no correlation has been observed between conserved amino acid residues/structures and EC50 values of interprotomer-targeting compounds (Abdelnabi et al., 2019, Plos Biol). As shown in Fig. 4D, the three residues provides conserved structures in enterovirus, but could not explain/predict the effectiveness of interprotomer-targeting compounds to enterovirus infection (e.g., EC50 values). The importance of identified interaction between the three amino acids and the compounds in the overall interaction, and also in the effectiveness might be discussed.
2. The correlation between EC50 or actual affinity of the compounds and estimated binding energy is not clear. The authors might explain this point. If necessary, new experiments to validate the implicated binding energy and actual affinity might be provided.
3. There seems apparent discrepancy between the effect of glutathione and that of interprotomer-targeting compounds. Glutathione is considered to be essential for virion assembly of some strains of enterovirus (Thibaut et al., 2014, Plos Pathog, Ma et al., 2014, Plos Pathog). The common mode of interaction shown in Fig. 4 might suggest interprotomer-targeting compounds could serve as an enhancer of virus infection in place of glutathione in glutathione-depleted condition. The authors might explain about this apparent discrepancy to evaluate the potency of interprotomer-targeting compounds.
4. Throughout the manuscript, the authors suggested the pros and cons of capsid binders, classical pocket-targeting capsid binders, and interprotomer-targeting compounds. However, some seem not to be supported by scientific evidences. For example, both classical pocket-targeting capsid binders, and interprotomer-targeting compounds could induce resistant viruses. EC50 values of interprotomer-targeting compounds are generally higher than those of classical pocket-targeting capsid binders. The authors might clarify the property of compounds.

Specific points:

1. The pros and cons should be discussed with proper original articles.

Summary: ... relatively low manufacturing costs, high oral bioavailability, and high shelf stability...

Abstract: efficacy, toxicity, drug-resistant viruses...

2. Chemical structures of CP17 and CP47 might be shown.

3. Fig.4D: E11 ?2233

Reviewer #3 (Remarks to the Author):

The manuscript by Flatt et al. identifies a conserved network within an interprotomer pocket responsible for virion stability in enteroviruses. Small molecule compounds CP17 and CP48 were shown to interact within an interprotomer pocket of CVB3 and CVB4 and were analyzed using single particle cryo-EM and in vitro thermal stability assays. The authors concluded that three residues that reside at the VP1-VP3 interface were conserved across multiple enterovirus species. The authors had previously identified the existence of this interprotomer pocket between VP1 and VP3 structural proteins by interaction of compound CP17 with CVB3. Through this initial work, the authors found three amino acids (VP1-Arg219, VP1-Arg234, and VP3-Gln233) responsible for playing a role in binding of CP17 and stabilizing the virion. They present reprocessed data from that original work and now report the complex at 2.8Å. The authors set out to examine the role of these sites in another virus CVB4 complexed with a compound similar to CP17. To validate that this related compound CP48 behaved similar to CP17, the authors carry out a thermal stability assay of CVB4-CP48 and showed that CP48 does increase viral stability at increased temperatures, as was previously shown with CVB3-CP17. They do a cryo-EM reconstruction of CVB4 complexed with CP48 yielding a 2.7Å structure, which has the compound bound to the same three amino acids that were previously identified in CVB3. Using this new structure, they propose that because of the conservation of this pocket and residues, small molecules that bind here, like CP17 and CP 48, they would be useful to develop as pan-enterovirus inhibitors. This is an important set of observations that could help for broad spectrum antiviral development.

Several points to address:

Figure 1 – it should be clear that this structure was previously published at a lower resolution. This current structure is important because of the ability to discern side chains and interaction of CP17.

Figure 3 – The authors claim that this is the first structure of CVB4 but do not show any data for the native structure other than in figure 3C. Can they provide the RMSD between native and CP48 bound? What is the resolution of the native structure? In 3C, they show density, can they model the pocket factor? How might CP48 influence binding if the pocket doesn't change conformation?

It is somewhat surprising that they didn't do structures of a single virus complexed with both compounds (i.e. CVB4 with CP17 or CP48).

In the Discussion (line 5) they mention "mutational capacity". Did they mutate any or all of the three conserved residues in CVB4 and show either lethality for the virus, or reduced CP48 binding?

Dear reviewers,

We thank you for taking the time to read the manuscript, and for providing helpful feedback. We believe now that the content of the manuscript has been greatly improved. Please find below our responses to your specific comments in blue.

Reviewer #1 (Remarks to the Author):

Review for COMMSBIO-20-2076-T

This manuscript by Flatt et al, "Identification of a conserved virion-stabilizing network inside the interprotomer pocket of enteroviruses" reports on two cryo-EM derived structures of icosahedral viruses bound to drug compounds. Enteroviruses are important human pathogens and an atomic level understanding of how drug compounds interact with viral proteins is an important area of research.

In this work, the authors reprocess old data (formerly a 4 Å map) to generate a map closer to 2.8 Å, allowing them to fully model the proteins and the bound ligand. In addition, they solved a novel structure to 2.7 Å. The work is largely structural, although they also do thermal titrations to show that the virions are stabilized against heat when bound to the compounds. I have very few issues with the quality of the data or the authors' main conclusions. The cryo-EM data collection and interpretation section is high quality, and is thorough. However, there are several points missing throughout the manuscript, and I have a number of suggestions regarding the presentation that would make this manuscript easier to digest.

The Introduction is generally well written and comprehensive. The term "protomer" could be confusing to non-experts and I suggest providing a better description of this term.

In the revised text, we describe a protomer in the following three places:

1) Opening sentence of paragraph 2 in the Introduction:

"CVB capsids share a common enteroviral architecture constructed from 60 repeating asymmetric units termed protomers, each consisting of the four structural proteins VP1, VP2, VP3, and VP4 (Jiang et al., 2014)."

2) Neighboring protomers are clearly labeled in Figure 1 B.

3) From the first paragraph of results for CP17 bound to the interprotomer pocket of CVB3:

"The interprotomer site is located between adjacent asymmetric units "protomers" in a narrow opening formed at the intersection of neighboring VP1 β -barrels and the C-terminus of a proximally situated VP3 molecule."

Suggestions on presentation:

The figures are hard to interpret. For example:

Figure 1: An enlarged view of the binding pocket with residue labels would help the reader immensely. Perhaps a "cage view" of the density would also help.

We appreciate your feedback and suggestions. Figure 1 has been revised to ease interpretation. We rearranged the panels, added labels for the pocket residues, and we included a ligand interaction diagram.

Figure 2: I assume the control was done at 49 and 52 and no active virus was detected? Can this be shown more clearly with an asterisk or a “ND” label for “not detected”?

You are correct. We clarify that this is the case in the last sentence of the legend for Figure 2: “No virus was detected in the control samples at 49 and 52 degrees Celsius at the TCID₅₀/ml detection limit of 2 log₁₀.”

Figure 3C: superimposing the two structures with different colors would help interpret the similarities and differences better. And the pink density blob is not particularly effective in showing the point. Lastly, a view showing residue labels would help quite a bit (same as for figure 1).

We made a new Figure 3 in which we compare CVB4-CP48 with the CVB4 control. We used your feedback and suggestions for Figure 1 when designing the layout. The old Figure 3 where we compare CVB3-CP17 and CVB4-CP48 is now Supplementary Figure 1. We revised the Supplementary Figure 1 so that the asymmetric units for CVB4+CP48 and CVB4 control are overlaid (Supplementary Figure 1C). Also, we modified the view of the density corresponding to the pocket factor to better highlight the differences we see inside the VP1 pocket when comparing CVB4+CP48 with the control (Supplementary Figure 1B). Residue labels were added too.

Figure 4A: label the residues and/or add arrows. It is really hard to see what the authors are trying to highlight.

We revised Figure 4, and the new version contains residue labels.

Table of cryo-EM data: Was a defocus value of -5 μm really used? Seems rather unconventional. Also, please show a raw micrograph of the new data (supplementary is fine).

Thank you for pointing this out, there was an error here. The defocus values were updated to reflect the settings we used at the microscope to collect data. We have included raw aligned micrographs for the two new datasets from our study (CVB4+CP48 and CVB4) in Supplementary Figure 3.

General comments on drug effectiveness:

Interpretation on page 9: “Also, the control CVB4 structure has no density filling the interprotomer pocket. The alteration may result from a change in the structural dynamics of the pocket factor, or because this region is an additional weak binding site for CP48”.....These structures were solved using icosahedral symmetry, so there could also be issues with stoichiometry and occupancy. If all 60 sites are not occupied, the signal would simply be averaged out.

We modified the sentence from paragraph 3 of the results for CP48 within the CVB4 virion:

“The alteration may result from a change in the structural dynamics of the pocket factor, if this region is an additional weak binding site for CP48, or it may reflect a substoichiometric occupancy issue where signal is averaged away by applying icosahedral symmetry during image processing.”

Related to the point above, on page 11, the authors state “When an inhibitor is stably anchored

to the network (60 sites per a single virion), it interferes with motion transmission such that the virus particle cannot undergo expansive conformational changes in the interprotomer region, and hence is unable to uncoat the genome at precisely the right time in infection.” Is it known if full occupancy is required for the drug to function? An experiment doing a titration of the compound along with virus infectivity assays. If this is known, please cite the previous work.

It is not known whether or not full occupancy is required for the drug to function. In the experiments where we titrate a known amount of compound with virus and then perform infectivity assays there are certain limitations that prevent us from addressing the question. These include that we are working with a crude solution where we do not know for sure if the compound binds only to virus. Also, the virus particle estimate for titration experiments is based on infectious units, but there are probably also non-infectious viruses present that could still bind the compound.

In the materials and methods the drug compounds were incubated with virus at 10mg/mL. This seems a very high concentration and I wonder how tight the binding really is. Is this dose toxic to humans? See above comment on titration... if this study really is evaluating the binding potential of these viruses, the nature of the binding really should be explored further. As one example, comprehensive site directed mutagenesis would be a logical follow up to determine which of the residues are critical for drug binding.

Thank you for carefully reading through the materials and methods and bringing this particular issue to our attention. The drug stock solution was 10 mg/mL, which is then diluted when added to the purified virus for cryoEM, but is still in a huge excess. The final concentration for CP48 in the cryo-sample was 0.17 mg/mL (440 uM), which gives a 2,500 molar excess of the capsid binder in comparison to the virus. We used an excess amount to ensure that each interprotomer site would be occupied. In this experiment we used the protein concentration for the molarity calculation.

We made the following correction to the methods section for cryo-EM sample preparation and data collection:

“The compound 4-[[6-propoxy-2-naphthyl)sulfonyl]amino}benzoic acid (CP48) was ordered from a commercial supplier (www.specs.net) and dissolved in DMSO at a concentration of 10 mg/mL. We further diluted (10x dilution) the compound in a gradient/exchange buffer. Purified CVB4 and CP48 were then mixed at a molar ratio of 1:2500, which yielded a final capsid binder concentration of 0.17 mg/mL. The mixture was incubated at 37°C for 1 hour.”

CP48 toxicity was tested in BGM, Vero, and HeLa cell lines with a determined CC50 between 64 and 197 uM (Abdelnabi et al 2019). The antiviral activity of the compound is detected at much lower concentrations with EC50 values of approximately 8 uM for both CVB3 and CVB4 (Abdelnabi et al 2019).”

Site-directed mutagenesis has been carried out to analyze the CP17 interaction inside the CV3 Nancy interprotomer pocket (Abdelnabi et al. 2019). Specifically, 12 reverse-engineered mutants were generated in a CVB3 Nancy infectious clone. From the 12 mutants, 8 proved viable, and 7 led to resistance. The 4 that were nonviable included VP1 Q160G, VP1 R234G, VP3 F236G and VP3 Q233G. VP1 R219 was not tested. Unfortunately, we do not have an infectious clone for CVB4. We performed a sequence alignment of the region of VP1 that includes the two Arg residues for the 74 isolates of CVB4 in GenBank and found that they are both strictly conserved.

We revised the discussion to include information about site-directed mutagenesis:

“Structure-guided in vitro assays involving CVB3 and CP17 indicated that this class of capsid binders increases particle stability, which we have observed to be the case here for CVB4 and CP48. Also, in our previous study, reverse engineering mutation experiments revealed that four interprotomer mutants in CVB3 Nancy were not viable: VP1_Q160G, VP1_R234G, VP3 F236G, and VP3 Q233G (VP1 R219 was not tested) (Abdelnabi et al, 2019). We were unable to perform similar experiments with CVB4 here because there is no infectious clone available. Nevertheless, structural alignments and experimental data suggest a conserved virion-stabilizing network within the interprotomer pocket that is less tolerant to mutations, a promising result for efforts to develop antivirals.”

Reviewer #2 (Remarks to the Author):

This is a follow-up report of a recent article on interprotomer-targeting compounds to enterovirus by the authors (Abdelnabi et al., 2019, Plos Biol). Novel information provided in current study is 1) a high resolution structure of a CP17-CVB3 complex, and 2) a novel structure of a CP47-CVB4 complex. These structures revealed conserved three amino acid residues of enterovirus capsid proteins interacting with interprotomer-targeting compounds.

The direct importance suggested by this manuscript might be as follows:

1. Mode of interaction between interprotomer-targeting compounds and critical amino acid residues (two Arg residues and one Gln residue), which is similar to a glutathione-EVF3 complex (Duyvesteyn et al., 2020, Commun Biol).
2. Atomic structure of CVB4.

A number of structures of enterovirus have been solved by atomic level, including the structures of interprotomer pocket. With identified critical amino acid residues, structure-guided drug design might provide more potent interprotomer-targeting compounds.

Atomic structure of CVB4 could provide a structural basis for understanding of CVB4 infection and for development of potent inhibitors.

Some parts of this manuscript might be clarified:

1. Apparently, no correlation has been observed between conserved amino acid residues/structures and EC50 values of interprotomer-targeting compounds (Abdelnabi et al., 2019, Plos Biol). As shown in Fig. 4D, the three residues provides conserved structures in enterovirus, but could not explain/predict the effectiveness of interprotomer-targeting compounds to enterovirus infection (e.g., EC50 values). The importance of identified interaction between the three amino acids and the compounds in the overall interaction, and also in the effectiveness might be discussed.

Thank you for raising a very important point. Beyond the three conserved residues, there are other amino acids lining the pocket that influence binding and thus effectiveness. To show a more complete view of these various contributions we have now included ligand interaction diagrams in Figures 1 and 3. The diagrams capture the full context, by showing the residues in contact with the compounds, and they include information about the nature of the contacts (charged, salt bridge, pi-pi stacking, etc.). Also, Figure 4 has been revised in response to the point you have raised. Specifically, we expanded the information to include important residues outside of the highly conserved three at the core, and in Figure 4C we look at conservation among the available enterovirus/rhinovirus structures from the PDB. We believe that these

changes, along with the changes to the text (see below), help provide a sense for why the effectiveness of interprotomer-targeting compounds varies, depending on the structure of the small molecule and the virion pocket.

We modified text in the following places:

From the results section for CP17 bound to the interprotomer pocket of CVB3 we added: “In addition, there are other contributions inside the pocket such as a cysteine residue in VP1 (C73) and hydrophobic residues (VP1 F76 and VP3 F236) that interact with the benzene rings of the compound, contributing to the overall binding energy and specificity of CP17 (Figure 1C and 1E).”

From the results section for CP48 within the CVB4 virion we added:

“Similar to how CP17 is anchored to CVB3, CP48 is stabilized by stacking interactions with two hydrophobic residues, Y67 in VP1 and F236 in VP3 (Figure 3C and 3D). The C64 residue in VP1, though not involved in the direct interaction with CP48, is in close vicinity to the compound (Figure 3D).”

We updated a header from “Three key residues form a virion-stabilizing contact network in the presence of binders” to “The virion-stabilizing network inside the pocket”

We also revised the section “The virion-stabilizing network inside the pocket”:

“The network is comprised of about 15 residues with 3 highly conserved amino acids forming the core of the binding site. These three residues, each from a different polypeptide chain, dictate the size and shape of interprotomer-targeting compounds, as well as the mechanism of action (Figure 4A). Other interactions that define the structure-activity relationship within the pocket include a pair of hydrophobic residues that stably position the benzene scaffolds of the inhibitors, and a cysteine residue that can hydrogen bond with the carboxylic end (Figure 4 B and C).”

2. The correlation between EC50 or actual affinity of the compounds and estimated binding energy is not clear. The authors might explain this point. If necessary, new experiments to validate the implicated binding energy and actual affinity might be provided.

In the current form of the manuscript, we first show that our predicted binding energies for CP17 and CP48 are in line with what has been reported for potent capsid binders that target the VP1 hydrophobic pocket. Then when comparing the binding energies for CP17 vs CP48 we specify the measured EC50 values from our previous work (Abdelnabi et al., 2019) as it may help to contextualize the differences in EC50s for the two binders. However, we believe that to take it a step further and make the robust correlation would require that we analyze the experimental structures of more than just the two virus-inhibitor complexes we present here, and preferably at higher resolutions than we currently report. This work is ongoing in the lab.

3. There seems apparent discrepancy between the effect of glutathione and that of interprotomer-targeting compounds. Glutathione is considered to be essential for virion assembly of some strains of enterovirus (Thibaut et al., 2014, Plos Pathog, Ma et al., 2014, Plos Pathog). The common mode of interaction shown in Fig. 4 might suggest interprotomer-targeting compounds could serve as an enhancer of virus infection in place of glutathione in

glutathione-depleted condition. The authors might explain about this apparent discrepancy to evaluate the potency of interprotomer-targeting compounds.

In our previous study we only observed an effect if the compounds were added before infection (Abdelnabi et al. 2019). Also, we saw that the antiviral activity of CP17 was not affected by increased intracellular glutathione levels, which is consistent with inhibition being at an early stage of entry before virus assembly in the cytoplasm. Our thermostability data clarifies further that these compounds prevent uncoating. It seems thus unlikely that CP48 and CP17 bind isolated protomers to either accelerate or decelerate assembly. This is in contrast to pleconaril which, in only binding to VP1, can also block the infectivity of progeny viruses (Zhang, Y., A. A. Simpson, R. M. Ledford, C. M. Bator, S. Chakravarty, G. A. Skochko, T. M. Demenczuk, A. Watanyar, D. C. Pevear, and M. G. Ross-mann. 2004. Structural and virological studies of the stages of virus replication that are affected by antirhinovirus compounds. *J. Virol.* 78:11061–11069)

In light of your comment we have revised the end of the discussion:

“Interestingly, Duyvesteyn et al. recently published a 1.8 Å resolution X-ray structure of bovine enterovirus F3 (EV-F3) with glutathione (GSH) positioned in a similar way within the pocket (Duyvesteyn et al., 2020). The antioxidant engages the same virion-stabilizing network as CP17 and CP48, which is not surprising given that these molecules have strikingly similar geometrical and chemical features (Supplementary Figure 2). Specifically, GSH adopts a hook-shaped structure with a carboxylic end and a sulfur-containing elbow region. The overall size approximates that of inhibitors that occupy the interprotomer site. The carboxylic group interacts with the two Arg residues from neighboring VP1 polypeptides inside the pocket, while the sulfur atom interacts with the oxygen in the Gln side chain of VP3. It is believed that for CVB3 and CVB4, GSH makes strong interactions with adjacent protomers to facilitate intracellular assembly of progeny virions (Smith and Dawson, 2006, Ma et al., 2014, Thibaut et al., 2014). Further work is necessary to understand how these molecules, which share similar shape and chemistry, modulate stability at the pocket to either prevent uncoating or facilitate assembly. Outside of structural efforts, it will be important to assess the influence of cellular cues and factors on their different modes of action. Accordingly, we foresee the development of improved binders, as well as an enhanced understanding of the biological significance of the interprotomer site, by building on the information provided by these new structures which show molecules inside the pocket.”

4. Throughout the manuscript, the authors suggested the pros and cons of capsid binders, classical pocket-targeting capsid binders, and interprotomer-targeting compounds. However, some seem not to be supported by scientific evidences. For example, both classical pocket-targeting capsid binders, and interprotomer-targeting compounds could induce resistant viruses. EC50 values of interprotomer-targeting compounds are generally higher than those of classical pocket-targeting capsid binders. The authors might clarify the property of compounds.

In the Discussion, we now refer to an example study from Pevear *et al* that was a clinical study on the use of pleconaril including EC50 values but also includes a nice evaluation of susceptibility of clinical isolates to pleconaril. In addition, we provide a reference for the isolation of lower susceptibility variants to pleconaril (Groarke and Pevear, 1999) and point out that such variants can also be isolated for CP17.

The first paragraph of the Discussion now reads “Despite decades of research on WIN antiviral compounds with e.g. EC50 value for pleconaril against human rhinovirus B1 reported as 0.2 µM, no drugs have been approved for use against enteroviruses²⁶. Recently, a new class of broad-

spectrum capsid binders was described, which inhibit a variety of enteroviruses by occupying a positively-charged surface depression in the interprotomer zone with e.g. EC50 value for CP17 against CVB3 as 0.7 μ M. Structure-guided in vitro assays involving CVB3 and CP17 indicated that this class of capsid binders increases particle stability, which we have observed to be the case here for CVB4 and CP48. Virus variants with reduced susceptibilities to compounds targeting either pocket can be selected under pressure, with concomitant reduced viability 21,27. Reverse engineering mutation experiments revealed that four interprotomer mutants in CVB3 Nancy were not viable: VP1 Q160G, VP1 R234G, VP3 F236G, and VP3 Q233G (VP1 R219 was not tested) 21. We were unable to perform similar experiments with CVB4 here because there is no infectious clone available. Nevertheless, structural alignments and experimental data suggest a conserved virion-stabilizing network within the interprotomer pocket that is less tolerant to mutations, a promising result for efforts to develop antivirals..”

Specific points:

1. The pros and cons should be discussed with proper original articles.

Summary: ... relatively low manufacturing costs, high oral bioavailability, and high shelf stability...

We removed the sentence about low manufacturing costs, high oral bioavailability, and high shelf stability from the summary as it is not our aim to go into a lengthy discussion on this particular aspect of antiviral development.

The updated sentence now reads:

“Small molecule capsid binders have the potential to be developed as antivirals that prevent virus attachment and entry into host cells.”

Abstract: efficacy, toxicity, drug-resistant viruses... 2. Chemical structures of CP17 and CP47 might be shown.

We have included the following reference to the introduction regarding efficacy, toxicity, and drug-resistant viruses, which goes along with Thibaut et al., 2012:

Antivirals against enteroviruses: a critical review from a public health perspective. *Antiviral Therapy* 2015; 20:121-130

Chemical structures for the compounds are shown in the new Figure 1E and Figure 3D.

3. Fig.4D: E11 ?2233

Figure 4 has been corrected.

Reviewer #3 (Remarks to the Author):

The manuscript by Flatt et al. identifies a conserved network within an interprotomer pocket responsible for virion stability in enteroviruses. Small molecule compounds CP17 and CP48 were shown to interact within an interprotomer pocket of CVB3 and CVB4 and were analyzed using single particle cryo-EM and in vitro thermal stability assays. The authors concluded that three residues that reside at the VP1-VP3 interface were conserved across multiple enterovirus species. The authors had previously identified the existence of this interprotomer pocket between VP1 and VP3 structural proteins by interaction of compound CP17 with CVB3. Through this initial work, the authors found three amino acids (VP1-Arg219, VP1-Arg234, and VP3-Gln233) responsible for playing a role in binding of CP17 and stabilizing the virion. They present reprocessed data from that original work and now report the complex at 2.8Å. The

authors set out to examine the role of these sites in another virus CVB4 complexed with a compound similar to CP17. To validate that this related compound CP48 behaved similar to CP17, the authors carry out a thermal stability assay of CVB4-CP48 and showed that CP48 does increase viral stability at increased temperatures, as was previously shown with CVB3-CP17. They do a cryo-EM reconstruction of CVB4 complexed with CP48 yielding a 2.7Å structure, which has the compound bound to the same three amino acids that were previously identified in CVB3. Using this new structure, they propose that because of the conservation of this pocket and residues, small molecules that bind here, like CP17 and CP 48, they would be useful to develop as pan-enterovirus inhibitors. This is an important set of observations that could help for broad spectrum antiviral development.

Several points to address:

Figure 1 – it should be clear that this structure was previously published at a lower resolution. This current structure is important because of the ability to discern side chains and interaction of CP17.

We added the following to the Figure 1 legend:

“The map was resolved to 2.8 Å upon reprocessing raw data from a previous publication (Abdelnabi et al., 2019)”

Figure 3 – The authors claim that this is the first structure of CVB4 but do not show any data for the native structure other than in figure 3C. Can they provide the RMSD between native and CP48 bound? What is the resolution of the native structure? In 3C, they show density, can they model the pocket factor? How might CP48 influence binding if the pocket doesn't change conformation?

Thank you for the comment. In response, we made a new Figure 3 where we compare CVB4+CP48 to the CVB4 control. In the new figure we show an enlarged portion of the cryoEM density map in 3A, and then we do a pocket comparison in 3C. A revised version of the old Figure 3 is now Supplementary Figure 1.

It is somewhat surprising that they didn't do structures of a single virus complexed with both compounds (i.e. CVB4 with CP17 or CP48).

Currently, we do not have any CP17 and only a limited amount of CP48, which we are using in new structural experiments involving other enteroviruses. However, we believe that this manuscript is a solid piece to be published now, and we will continue to study and publish on interprotomer-targeting compounds in the future.

In the Discussion (line 5) they mention “mutational capacity”. Did they mutate any or all of the three conserved residues in CVB4 and show either lethality for the virus, or reduced CP48 binding?

Site-directed mutagenesis experiments were done for CVB3+CP17 in our previous paper (Abdelnabi et al, 2019). Twelve residues in total were tested. Reverse-engineered CVB3 mutants VP3 Q233G and VP1 R234G were nonviable. The other conserved R219 was not tested. For CVB4, we are not able to do such experiments as there is no infectious clone.

We updated the discussion as follows:

“Structure-guided in vitro assays involving CVB3 and CP17 indicated that this class of capsid binders increases particle stability, which we have observed to be the case here for CVB4 and CP48. Virus variants with reduced susceptibilities to compounds targeting either pocket can be selected under pressure, with concomitant reduced viability 21,27. Reverse engineering mutation experiments revealed that four interprotomer mutants in CVB3 Nancy were not viable: VP1 Q160G, VP1 R234G, VP3 F236G, and VP3 Q233G (VP1 R219 was not tested) 21. We were unable to perform similar experiments with CVB4 here because there is no infectious clone available. Nevertheless, structural alignments and experimental data suggest a conserved virion-stabilizing network within the interprotomer pocket that is less tolerant to mutations, a promising result for efforts to develop antivirals.”

REVIEWERS' COMMENTS:

Reviewer #2 (Remarks to the Author):

Thank you very much for kindly answering to my concerns.
I agree with that answering to some concerns might be beyond current focus of study as described.

Minetaro Arita

Reviewer #3 (Remarks to the Author):

The authors of this manuscript describe an interprotomer pocket on the virion that is important for Coxsackieviruses B3 and B4 stability. Using small molecule inhibitors they demonstrate binding as visualized by cryoEM for CVB4. They have responded in a satisfactory manner to the comments of the three reviewers and I am satisfied that the manuscript has been significantly improved. The work represents an important finding for antivirals targeted at enteroviruses.